# No evidence for an association of plasma homocysteine levels and refractive error – Results from the population-based Gutenberg Health Study (GHS)

Stefan Nickels[1☯]*, Henk J. Blom[2,3☯], Andreas Schulz[4], Lutz Joachimsen[5], Thomas Münzel[6,7], Philipp S. Wild[6,7,8,9], Manfred E. Beutel[10], Maria Blettner[11], Karl J. Lackner[12], Norbert Pfeiffer[1‡], Wolf A. Lagrèze[5‡]

1 Department of Ophthalmology, University Medical Center of the Johannes Gutenberg-University Mainz, Mainz, Germany, 2 Laboratory of Clinical Biochemistry and Metabolism, Department of General Pediatrics, Adolescent Medicine and Neonatology, University Medical Centre Freiburg, Freiburg, Germany, 3 Metabolic Unit, Department of Clinical Genetics, Center for Lysosomal and Metabolic Diseases, ErasmusMC, Rotterdam, The Netherlands, 4 Preventive Cardiology and Preventive Medicine/Center for Cardiology, University Medical Center of the Johannes Gutenberg-University Mainz, Mainz, Germany, 5 Eye Center, Medical Center, Faculty of Medicine, University of Freiburg, Freiburg, Germany, 6 Center for Cardiology I, University Medical Center of the Johannes Gutenberg-University Mainz, Mainz, Germany, 7 Center for Translational Vascular Biology (CTVB), University Medical Center of the Johannes Gutenberg-University Mainz, Mainz, Germany, 8 Center for Thrombosis and Hemostasis (CTH), University Medical Center of the Johannes Gutenberg-University Mainz, Mainz, Germany, 9 German Center for Cardiovascular Research (DZHK), partner site Rhine-Main, Mainz, Germany, 10 Department of Psychosomatic Medicine and Psychotherapy, University Medical Center of the Johannes Gutenberg-University Mainz, Mainz, Germany, 11 Institute for Medical Biostatistics, Epidemiology and Informatics, University Medical Center of the Johannes Gutenberg-University Mainz, Mainz, Germany, 12 Institute for Clinical Chemistry and Laboratory Medicine, University Medical Center of the Johannes Gutenberg-University Mainz, Mainz, Germany

☯ These authors contributed equally to this work.
‡ These authors also contributed equally to this work.
* post@stefan-nickels.de

**Data Availability Statement:** The written informed consent of GHS study participants does not approve public access to the data. This concept

## Abstract

### Purpose

There is a strong association between severe hyperhomocysteinemia and myopia. Thus we studied the hypothesis that even moderately increased levels of homocysteine (Hcy) might be a potentially treatable risk factor for myopia.

### Methods

The Gutenberg Health Study (GHS) is a population-based, prospective, observational cohort study in Germany, including 15,010 participants aged between 35 and 74 at recruitment. The baseline examination was conducted from 2007–2012. Refraction was measured using autorefraction (HARK 599, Carl Zeiss AG, Jena, Germany). Hcy was measured by an immunoassay. We included only phakic participants without a history of corneal surgery or corneal laser treatment. We used linear regression models to evaluate the potential association between Hcy and refraction at baseline, and between Hcy and change in refraction

was requested by the local data protection officer and ethics committee (local ethics committee of the Medical Chamber of Rhineland-Palatinate, Germany). Access to data at the local database in accordance with the ethics vote is offered upon request at any time. Interested researchers can make their requests to the Principal Investigators of the Gutenberg Health Study (email: info@ghs-mainz.de).

**Funding:** The Gutenberg Health Study is funded through the government of Rhineland-Palatinate ("Stiftung Rheinland-Pfalz für Innovation", contract AZ 961-386261/733), the research programs "Wissen schafft Zukunft" and "Center for Translational Vascular Biology (CTVB)" of the Johannes Gutenberg-University of Mainz, and its contracts with Boehringer Ingelheim, and PHILIPS Medical Systems, including an unrestricted grant for the Gutenberg Health Study. Philipp S. Wild is funded by the Federal Ministry of Education and Research (BMBF 01EO1503) and he is PI of the German Center for Cardiovascular Research (DZHK). Henk J Blom received a research grant of Orphan Europe on homocystinuria and myopia." The funders had no role in study design, data collection and analysis, decision to publish, or preparation of the manuscript.

**Competing interests:** The authors have read the journal's policy and have the following potential competing interests to declare: the Gutenberg Health Study is funded in part through contracts with Boehringer Ingelheim, and PHILIPS Medical Systems. This does not alter our adherence to PLOS ONE policies on sharing data and materials.

between baseline and 5-year-follow-up examination. We used generalized estimating equation models to account for the correlation between fellow eyes.

## Results

We included 13,749 participants, categorized as having no myopia (spherical equivalent > -0.75 D, 65.2%), low myopia (-0.75 D–-2.75 D, 21.5%), moderate myopia (-3.00 D– 5.75 D, 9.8%) and high myopia ($\leq$ -6 D, 3.5%). Median Hcy levels were similar in all groups (μmol/l). We observed no association of Hcy with refraction or 5-year change in refraction in the models adjusted for age, sex and socioeconomic status.

## Conclusion

We found no evidence for an association of Hcy levels and refractive error.

## Introduction

Myopia is a worldwide increasing health challenge not only requiring optical aids such as glasses or contact lenses but also increasing the risk of severe secondary eye diseases such as glaucoma, cataract and macular degeneration [1]. Both hereditary and environmental risk factors have been identified or suspected [2]. In this exploratory study, we aim to evaluate whether elevated levels of homocysteine (Hcy) might be an additional risk factor for the development of myopia. The normal level of plasma Hcy is 5–15 μmol/l, and therapy is usually initiated if plasma Hcy is above 50μmol/l [3]. Patients with classical hyperhomocysteinemia typically present with myopia and if untreated even with ectopia lentis. Other symptoms that may occur are mental retardation, marfanoid habitus, osteoporosis, thromboembolic events and behavioral deficits [4]. Milder forms exist and may manifest late in adulthood with myopia or ectopia lentis as a first sign while others may even remain asymptomatic for many years [5, 6]. In a cohort of patients with late-diagnosed hyperhomocysteinemia, all 14 had lens subluxation or dislocation at diagnosis [7]. Some of them attended ophthalmologic care for many years before diagnosed with hyperhomocysteinemia because of lens subluxation and progressive myopia. All patients with poor biochemical control were myopic. Given the very strong association between severe hyperhomocysteinemia and myopia, we hypothesized that even moderately increased levels of Hcy might be a risk factor for myopia [3, 8]. We assume that moderately increased Hcy would be present over the entire life span and could thus be detected in our adult study sample.

## Materials and methods

### Study population

The Gutenberg Health Study (GHS) is an ongoing population-based, prospective, single-center cohort study at the medical center of the Johannes Gutenberg University Mainz in Germany [9]. The sample was randomly drawn and equally stratified by sex and area of residence for each decade of age. Exclusion criteria were insufficient knowledge of German and physical or mental inability to participate in the examinations in the study center. The study protocol and study documents were approved by the local ethics committee of the Medical Chamber of Rhineland-Palatinate, Germany. According to the tenets of the Declaration of Helsinki, written informed consent was obtained from all participants prior to entering the study. The

baseline examination with a total of 15,010 participants aged 35 to 74 years took place from 2007 to 2012 and comprised an ophthalmic examination, several general and cardiovascular examinations, as well as interviews, and questionnaires [10]. The five-year-follow-up started in April 2012 and was finished in April 2017. For the analysis of refraction, we included all participants with available information on Hcy and refraction at baseline and with phakic lens status and no history of corneal surgery or corneal laser treatment. For the analysis of 5-year change in refraction, we further excluded all participants without Hcy or refraction measurements at the 5-year follow-up examination and with lens replacement surgery, corneal surgery or corneal laser treatment between baseline and follow-up examination.

## Homocysteine measurements

Peripheral venous blood samples were obtained according to standard operation procedures, using a sampling container with a Hcy-stabilizing additive ("S-Monovette® HCY-Z-Gel", Sarstedt AG & Co, Nürnberg, Germany), and were subsequently transported on ice to the main laboratory. After cooled centrifugation homocysteine was measured in plasma by immunoassay on an Architect i2000SR analyzer (Abbott Diagnostics, Wiesbaden, Germany). We categorized Hcy levels into normal ($< = 15$ μmol/l), mild hyperhomocysteinemia ($>15$ to $< = 30$ μmol/l), intermediate hyperhomocysteinemia ($> 30$ to $< = 100$ μmol/l), and serious hyperhomocysteinemia ($> 100$ μmol/l) [11].

## Ophthalmologic parameters

The ocular characteristics were obtained during the ophthalmological examination as described elsewhere [10]. In brief, non-cycloplegic refraction and best-corrected distance visual acuity were measured in both eyes, starting with the right eye, using a Humphrey Automated Refractor / Keratometer (HARK) 599 (Carl Zeiss AG, Jena, Germany). Visual acuity was measured using the built-in Snellen charts, ranging from 20/400 to 40/20 (decimal 0.05 to 2.0). For lower levels of visual acuity, we used a visual acuity chart at a distance of one meter up to 20/800, and then counting fingers, hand movements, and test of light perception. The spherical equivalent (SE) was calculated as the spherical correction value plus half the cylindrical power. Participants were then categorized by their SE into having low myopia ($< = -0.5$ D —>-6 D in either eye), high myopia ($< = -6D$ in either eye), or no myopia ($>-0.5$ D in both eyes), following the recent International Myopia Institute (IMI) recommendation [12]. Intraocular pressure (IOP) was measured with an air-puff noncontact tonometer (Nidek NT-2000; Nidek, Co., Gamagori, Japan). Starting with the right eye, the mean of three measurements within a 3-mmHg range was obtained for each eye. Information on lens status and previous eye surgery was obtained from the medical history collected preceding the ophthalmological examination. Self-reported lens status at baseline was validated by slit-lamp examination [13].

## Socio-demographic characteristics and comorbidities

During a computer-assisted personal interview participants were asked about their medical history and income, school education, vocational training, and occupational status. The socioeconomic status (SES) was defined based on income, education and position according to the SES-index used within the German Health Update 2009 (GEDA), with a range from 3 to 21 (3 indicates the lowest SES and 21 the highest SES) [14]. Diabetes mellitus was defined by fulfilling one of the following criteria: diabetes mellitus diagnosed by a physician, known therapy (oral medication or insulin), or HbA1c $> = 6.5\%$. Dyslipidemia was defined by a low-density lipoprotein (LDL) to high-density lipoprotein ratio (LDL/HDL) of $>3.5$, lipid-lowering medication, or diagnosis by a physician. Hypertension was defined by the use of antihypertensive

medication, mean systolic blood pressure $\geq$ 140 mm Hg or mean diastolic blood pressure $\geq$ 90 mm Hg in 3 consecutive measurements at rest, or diagnosis of arterial hypertension by physician. Smoking was dichotomized into current smokers and non-smokers (including past smokers). Obesity was defined as a BMI $> =$ 30 m$^2$/kg.

### Statistical analysis

For continuous variables, we calculated median and 25$^{th}$ and 75$^{th}$ percentiles, and mean and standard deviation for approximately normal distributed variables. We used linear regression models with general estimating equations for consideration of the correlation between fellow eyes to assess the association of Hcy and refraction and change in refraction [15]. Model 1 included no covariates; model 2 was adjusted age, sex and socio-economic status. For the identification of potential confounders we followed the directed acyclic graph (DAG) concept [16]. We used the online tool DAGitty V2.3 (http://dagitty.net/, last accessed 2019-08-16) to visualize potential causal relationships (S1 Fig) and to derive the minimal sufficient adjustment set for estimating the total effect of Hcy on refraction [17]. As sensitivity analyses, we additionally calculated the regression models split my sex, limited to participants with Hcy >15 µmol/l, and limited to participants without severe astigmatism (Cyl > -1 diopter in both eyes). Due to the exploratory character of this study, we did not adjust for multiple testing. P-values should be interpreted with caution and in connection with effect estimates. We used R version 3.5.2 (2018-12-20) for the analysis [18].

### Results

After exclusion of 386 participants without refraction, 114 without homocysteine measurements, 677 with pseudophakia, and 84 with previous refractive surgery, our analysis of Hcy and refraction was based on 13,749 subjects (Table 1). 63% (n = 8,609) had no myopia, 33% (n = 4,540) had low myopia, and 4% (n = 600) had high myopia. 86% (n = 11,814) had normal Hcy levels ($< =$ 15 µmol/l), 13% (n = 1840) mild hyperhomocysteinemia (>15 to $< =$ 30 µmol/l), 0.7% (n = 91) intermediate hyperhomocysteinemia ($>$ 30 to $< =$ 100 µmol/l), and n = 4 serious hyperhomocysteinemia ($>$ 100 µmol/l).

Homocysteine levels were lower in women (median 10.1 µmol/l, 25$^{th}$ and 75$^{th}$ percentiles 8.50, 12.07) than in men (median 12.0 µmol/l, 10.30, 14.30).

The mean spherical equivalent was similar across categories of Hcy levels, and we did not observe a relation of Hcy levels and refraction (Figs 1 and 2), and the distribution of low and high myopia was similar as well (S2 Fig). In Hcy levels higher than 100 µmol/l, there are only few observations (n = 4). In the univariate linear regression analysis, a 10 µmol/l higher Hcy level was associated with a 0.17 D more myopic refraction (95% confidence interval 0.10–0.23). After adjusting for age, sex and socio-economic status, this association was no longer present (Table 2).

Our analysis of 5-year change in refraction was based on 9,928 participants (S1 Table). The mean change in refraction was 0.09 diopters (standard deviation 0.63). Again, we found no evidence for an association with Hcy, neither in the plots (S3 and S4 Figs), nor in the univariate and the adjusted regression analysis (Table 3).

The analyses stratified for men and women, as well as restricted to participants with homocysteinemia >15 µmol/l, and restricted to participants without severe astigmatism (both eyes cylinder > -1 diopter) revealed also no association (S3 Table).

**Table 1. Characteristics of the baseline sample of the German population-based Gutenberg Health Study (GHS), 2007–2012.**

|  | Overall (n = 13,749) | Men (n = 6975, 50.7%) | Women (6774, 49.3%) |
|---|---|---|---|
| Age [years] | 54.42 (10.93) | 54.71 (10.95) | 54.12 (10.89) |
| Socio-economic status | 13.00 [9.00, 17.00] | 13.50 [10.00, 18.00] | 12.00 [9.00, 15.00] |
| Hypertension | 6701 (48.8) | 3749 (53.8) | 2952 (43.6) |
| Diabetes mellitus | 1203 (8.7) | 750 (10.8) | 453 (6.7) |
| Dyslipidemia | 4693 (34.2) | 2986 (42.9) | 1707 (25.2) |
| Obesity (BMI> = 30) | 3404 (24.8) | 1802 (25.8) | 1602 (23.7) |
| Smoking | 2717 (19.8) | 1480 (21.3) | 1237 (18.3) |
| Self-reported cancer | 1184 (8.6) | 535 (7.7) | 649 (9.6) |
| Sphere (OD) [diopter] | -0.12 (2.50) | -0.13 (2.41) | -0.11 (2.59) |
| Sphere (OS) [diopter] | -0.12 (2.49) | -0.13 (2.43) | -0.10 (2.55) |
| Cylinder (OD) [diopter] | -0.57 (0.66) | -0.59 (0.68) | -0.54 (0.64) |
| Cylinder (OS) [diopter] | -0.56 (0.65) | -0.58 (0.66) | -0.54 (0.63) |
| Spherical equivalent (OD) [diopter] | -0.41 (2.53) | -0.43 (2.44) | -0.38 (2.63) |
| Spherical equivalent (OS) [diopter] | -0.39 (2.52) | -0.42 (2.45) | -0.36 (2.59) |
| Visual acuity (OD) [logMAR] | 0.00 [0.00, 0.10] | 0.00 [0.00, 0.10] | 0.00 [0.00, 0.10] |
| Visual acuity (OS) [logMAR] | 0.00 [0.00, 0.10] | 0.00 [0.00, 0.10] | 0.00 [0.00, 0.10] |
| Homocysteine [µmol/l] | 11.10 [9.30, 13.30] | 12.00 [10.30, 14.30] | 10.10 [8.50, 12.07] |

For categorical variables: absolute (relative) frequencies; for continuous variables: mean (standard deviation), in case of skewed distribution median (25th/ 75th percentile).

## Discussion

We found no evidence for an association between Hcy and refraction or 5 year-change in refraction. To our knowledge, this is the first study that has evaluated a potential relationship of Hcy and myopia. We assumed that moderately increased Hcy would be present over the

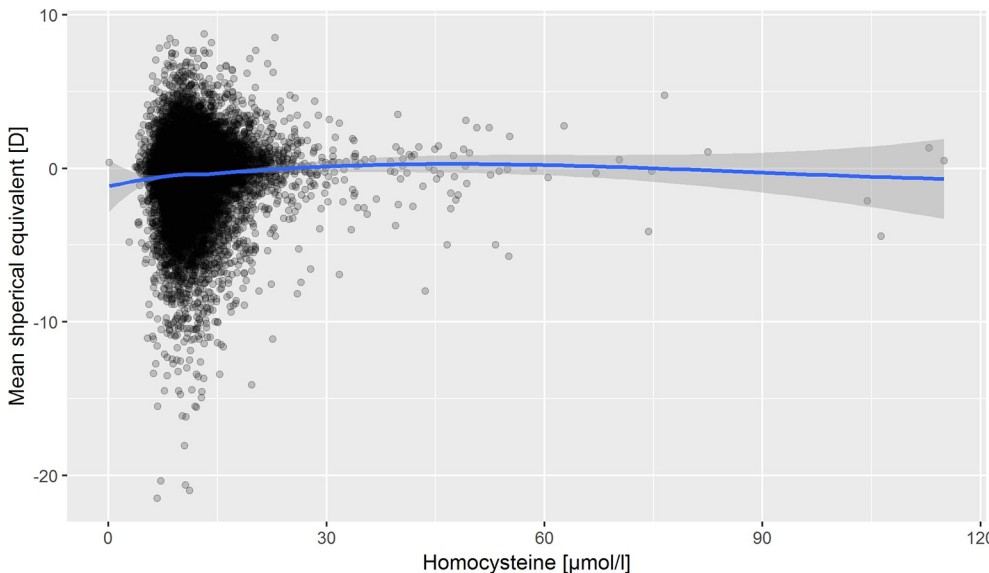

**Fig 1. The distribution of refraction in relation to homocysteine levels in the baseline sample of the German population-based Gutenberg Health Study (GHS), 2007–2012.** Smoothing line with 95% confidence bands based on locally weighted scatterplot smoothing (LOESS).

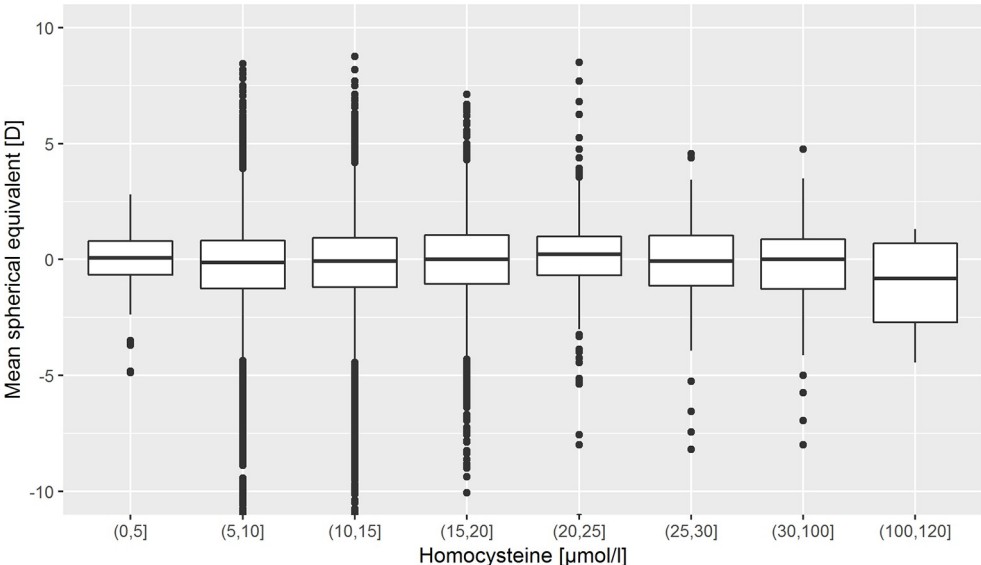

**Fig 2. Refraction in relation to grouped baseline homocysteine levels in the German population-based Gutenberg Health Study (GHS) 2007–2012.** Outliers more extreme than 11 diopters are not displayed. N included in homocysteine categories (left to right): 36, 4836, 6942, 1484, 272, 84, 91, 4.

entire life span and could thus be detected in our adult study sample. Given the proposed pathophysiological mechanism underlying the association of severe hyperhomocysteinemia and progressive myopia, namely that the extremely high Hcy concentration disrupts the zonular fibers, long-term exposure to moderately increased Hcy might also result in ocular effects like a more myopic refraction compared to the general population, and/or a continuous change towards more myopic refraction. This effect might either be observable cross-sectionally or in change over time–but we did not observe any effect in our study sample.

A limitation of our study is a lack in precision in the measurement of refraction by using autorefraction without cycloplegia. There is evidence from epidemiologic studies that the resulting misclassification is higher in children but not limited to this age group [19]. Furthermore, the misclassification is dependent on refractive error—our study outcome [20]. Therefore, there is the risk of a bias that might have diluted the effect we tried to explore. Despite this limitation, previous analyses e.g. on the relationship between refraction and education revealed that our measurement method is sufficiently precise to detect associations [21], but this does not necessarily be the case for the subject of current analysis. Another potential limitation could be the exclusion of 761 participants with a history of ocular surgery. By doing so, we might have excluded those participants with ectopia lentis due to high Hcy. On the other hand, our aim was to evaluate a potential influence of Hcy elevations below the threshold commonly used to initiate therapy, and these estimates would be unbiased by excluding severe

**Table 2. Association of homocysteine levels with diopters of spherical equivalent–Results from the German population-based Gutenberg Health Study (GHS).**

|  | Univariate (27,348 eyes) | | Adjusted for age, sex, and socio-economic status (27,348 eyes) | |
|---|---|---|---|---|
|  | Estimate (CI) | p-value | Estimate (CI) | p-value |
| Homocysteine [10 µmol/l] | 0.17 (0.10–0.23) | 6.58E-07 | -0.04 (-0.11–0.02) | 0.15 |

Results from linear regression, using generalized estimating equation models to account for the correlation of fellow eyes, p value derived by Wald score test.

**Table 3. Association of homocysteine levels with 5-year-change in diopters of spherical equivalent–Results from the German population-based Gutenberg Health Study (GHS).**

| | Univariate (19,768 eyes) | | Adjusted for age, sex, and socio-economic status (19,768 eyes) | |
|---|---|---|---|---|
| | Estimate (CI) | p-value | Estimate (CI) | p-value |
| Homocysteine [10 μmol/l] | -0.01 (-0.03–0.02) | 0.52 | -0.02 (-0.05–0.00) | 0.07 |

Results from linear regression, using generalized estimating equation models to account for the correlation of fellow eyes, p value derived by Wald score test.

cases of hyperhomocysteinemia. Hcy levels below this threshold are unlikely to influence the likelihood of participants to take part in the GHS, because there would be no effect on health and everyday life. Therefore we expect no selection bias. Hcy levels of participants might be influenced by the supplementary intake of vitamins. We were not able to account for vitamin status, intake of supplementary vitamins, and a history of hyperhomocysteinemia treatment in our analysis, because the data has not been collected. On the other hand, strengths of our study are the standardized sampling and examination program of the GHS ensuring a high data quality, and the large, population-based sample.

In summary, this is the first study to report on the potential association of Hcy levels and refraction in a large population-based sample. We found no evidence for an association of Hcy and refraction.

## Supporting information

**S1 Fig. Directed acyclic graph (DAG) to visualize causal relationships and to derive the minimal sufficient adjustment set for estimating the total effect of plasma homocysteine on refraction.** Generated with DAGitty 2.3 (http://dagitty.net/, last accessed 2019-08-19). SES = socio-economic status
(TIF)

**S2 Fig. Distribution of myopia categories by groups of homocysteine levels in the German population-based Gutenberg Health Study (GHS), 2007–2012.** Low myopia: spherical equivalent < = -0.5 D—>-6 D in either eye; high myopia: spherical equivalent < = -6 D in either eye, no myopia: spherical equivalent > -0.5 D in both eyes. N included in homocysteine categories (left to right): 36, 4836, 6942, 1484, 272, 84, 91, 4.
(TIF)

**S3 Fig. Change in refraction (comparing 5-year-follow-up with baseline) in relation to baseline homocysteine levels in the German population-based Gutenberg Health Study (GHS).** Smoothing line with 95% confidence bands based on locally weighted scatterplot smoothing (LOESS).
(TIF)

**S4 Fig. Change in refraction (comparing 5-year-follow-up with baseline) in relation to grouped baseline homocysteine levels in the German population-based Gutenberg Health Study (GHS).** N included in homocysteine categories (left to right): 36, 4836, 6942, 1484, 272, 84, 91, 4.
(TIF)

**S1 Table. Baseline characteristics of the German population-based Gutenberg Health Study (GHS) subsample for the analysis of 5-year change in refraction in relation to baseline homocysteine levels.**
(PDF)

**S2 Table. Distribution homocysteine levels by myopia categories in the baseline sample of the German population-based Gutenberg Health Study (GHS), 2007–2012.**
(PDF)

**S3 Table. Association of homocysteine levels (per 10 μmol/l]) with diopters of spherical equivalent (sensitivity analyses)–results from the German population-based Gutenberg Health Study (GHS).**
(PDF)

## Acknowledgments

We thank all study participants for their willingness to provide data for this research project and we are indebted to all coworkers for their enthusiastic commitment.

## Author Contributions

**Conceptualization:** Stefan Nickels, Henk J. Blom, Andreas Schulz, Thomas Münzel, Philipp S. Wild, Manfred E. Beutel, Maria Blettner, Karl J. Lackner, Norbert Pfeiffer, Wolf A. Lagrèze.

**Data curation:** Andreas Schulz.

**Formal analysis:** Stefan Nickels.

**Investigation:** Stefan Nickels, Henk J. Blom, Andreas Schulz.

**Project administration:** Stefan Nickels.

**Supervision:** Henk J. Blom, Philipp S. Wild, Maria Blettner, Norbert Pfeiffer.

**Visualization:** Stefan Nickels.

**Writing – original draft:** Stefan Nickels.

**Writing – review & editing:** Stefan Nickels, Henk J. Blom, Andreas Schulz, Lutz Joachimsen, Thomas Münzel, Philipp S. Wild, Manfred E. Beutel, Maria Blettner, Karl J. Lackner, Norbert Pfeiffer, Wolf A. Lagrèze.

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
