## [Decision Letter · Decision Letter 0]

14 Nov 2019

PONE-D-19-25431

No evidence for an association of plasma homocysteine levels and refractive error – Results from the population-based Gutenberg Health Study (GHS)

PLOS ONE

Dear Dr. Nickels,

Thank you for submitting your manuscript to PLOS ONE. After careful consideration, we feel that it has merit but does not fully meet PLOS ONE’s publication criteria as it currently stands. Therefore, we invite you to submit a revised version of the manuscript that addresses the points raised during the review process.

We would appreciate receiving your revised manuscript by Dec 29 2019 11:59PM. To enhance the reproducibility of your results, we recommend that if applicable you deposit your laboratory protocols in protocols.io, where a protocol can be assigned its own identifier (DOI) such that it can be cited independently in the future. For instructions see: http://journals.plos.org/plosone/s/submission-guidelines#loc-laboratory-protocols

We look forward to receiving your revised manuscript.

Kind regards,

Sabine Rohrmann

Academic Editor

PLOS ONE

Journal Requirements:

Reviewers' comments:

Reviewer's Responses to Questions

**Comments to the Author**

1. Is the manuscript technically sound, and do the data support the conclusions?

Reviewer #1: No

Reviewer #2: Yes

2. Has the statistical analysis been performed appropriately and rigorously? 

Reviewer #1: No

Reviewer #2: Yes

3. Have the authors made all data underlying the findings in their manuscript fully available?

Reviewer #1: No

Reviewer #2: Yes

4. Is the manuscript presented in an intelligible fashion and written in standard English?

Reviewer #1: Yes

Reviewer #2: Yes

5. Review Comments to the Author

Reviewer #1: The authors used a large longitudinal cohort comprising middle age to elderly subjects to test the hypothesis of elevated homocysteine levels as a risk factor for myopia and myopia progression. They reported no associated between the risk of interest and myopia or myopia progression.

The major concern of this study is the validation of hypothesis testing in this cohort.

Myopia onset and progression primarily happen in the children, particularly before age of 15 years. Furthermore, homocysteine levels increase with age. This means there is a problem in the temporary relationship between the risk (i.e. homocysteine) measurement and outcome in this study.

There is unlikely myopia progression after age of 35yr. The change of 0.09D with a large standard deviation for the 5-yr follow-up tells us essentially no change of refraction in this population. In addition, the change of refraction may be more likely due to the development of cataract rather than myopia progression.

SE contains both myopia and astigmatism components. Therefore, the authors need to exclude subjects with severe astigmatism.

Many other issues have been raised by the two reviewers.

Reviewer #2: The authors report about data of a potential link of Hcy levels with myopia in a large cohort. They found no evidence for an association of Hcy and refractive status of the participants.

The study shows an interesting topic, even as the authors present a very large cohort. Yet, some points should be considered. It is known that Hcy levels are dependent on the level of vitamin B6, B12, and folic acid. It would be of interest, if there were additional deficits in these levels. Where the patients treated for Hyper-Hcy in history (n, duration)?

I do not think figure S1 is necessary to show.

Dicussion: It would be of interest to discuss a the potential molecular basis of Hcy and myopia (e.g. LINE-1 hypermethylation), next to the limitations of the study. Additionally, it would be of interest the discuss the present data in the context of literature. Why do the authors think, that there is no association of Hcy and myopia – as there is evidence in literature for this association?

The GHS is a great study with an even large cohort of participants, thus the data are of great interest – yet, some minor points should be considered and the discussion could be improved.

6. PLOS authors have the option to publish the peer review history of their article (what does this mean?). If published, this will include your full peer review and any attached files.

Reviewer #1: No

Reviewer #2: No

---

## [Author Response · Author response to Decision Letter 0]

20 Feb 2020

As provided in the attached "response to reviewers" file:

Reviewer #1:

The authors used a large longitudinal cohort comprising middle age to elderly subjects to test the hypothesis of elevated homocysteine levels as a risk factor for myopia and myopia progression. They reported no associated between the risk of interest and myopia or myopia progression.

The major concern of this study is the validation of hypothesis testing in this cohort.

I am not sure if I understand the major concern of reviewer correctly. Despite stating a clear hypothesis of our exploratory analyses, namely “to evaluate whether elevated levels of homocysteine (Hcy) might be an additional risk factor for the development of myopia […] Given the very strong association between severe hyperhomocysteinemia and myopia, we hypothesized that even moderately increased levels of Hcy might be a risk factor for myopia.” (introduction), we did not apply the formal null hypothesis significance testing framework. 

Myopia onset and progression primarily happen in the children, particularly before age of 15 years. Furthermore, homocysteine levels increase with age. This means there is a problem in the temporary relationship between the risk (i.e. homocysteine) measurement and outcome in this study.

Our hypothesis is: “Given the very strong association between severe hyperhomocysteinemia and myopia, we hypothesized that even moderately increased levels of Hcy might be a risk factor for myopia” (introduction, line 77). We added the following sentence for clarification: “We assume that moderately increased Hcy would be present over the entire life span and could thus be detected in our adult study sample.”

There is unlikely myopia progression after age of 35yr. The change of 0.09D with a large standard deviation for the 5-yr follow-up tells us essentially no change of refraction in this population. In addition, the change of refraction may be more likely due to the development of cataract rather than myopia progression.

We do agree that myopia progression is unlikely after the age of 35 years in general. Assuming that moderately increased Hcy levels would result in a continuous progression of myopia, we might have been able to see this effect. We added the following sentences to the discussion to address this point: “We assumed that moderately increased Hcy would be present over the entire life span and could thus be detected in our adult study sample. Given the proposed pathophysiological mechanism underlying the association of severe hyperhomocysteinemia and progressive myopia, namely that the extremely high Hcy concentration disrupts the zonular fibers, long-term exposure to moderately increased Hcy might also result in ocular effects like a more myopic refraction compared to the general population, and/or a continuous change towards more myopic refraction. This effect might either be observable in a cross-sectionally or in change over time – but we did not observe any effect in our study sample.”

SE contains both myopia and astigmatism components. Therefore, the authors need to exclude subjects with severe astigmatism.

We addressed this point by adding a sensitivity analysis that is restricted to study participants with > -1 diopters cylinder in both eye. The estimates in this subsample without severe astigmatism did not substantially differ from the estimates of the whole analysis sample. We added the results to the supplement (S3 table).

Many other issues have been raised by the two reviewers.

We are thankful for the issues raised by the previous two reviewers, which resulted in an improved manuscript now subject to review. 

Reviewer #2:

The authors report about data of a potential link of Hcy levels with myopia in a large cohort. They found no evidence for an association of Hcy and refractive status of the participants.

The study shows an interesting topic, even as the authors present a very large cohort. Yet, some points should be considered. It is known that Hcy levels are dependent on the level of vitamin B6, B12, and folic acid. It would be of interest, if there were additional deficits in these levels. Where the patients treated for Hyper-Hcy in history (n, duration)?

The reviewer raised an important point. However, the main focus of the Gutenberg Health Study is to improve the risk stratification of cardiovascular disease. Therefore, not all variables relevant for special secondary research questions have been collected. We have now information regarding vitamin status, intake of vitamin supplements or a history of Hyper-Hcy treatment. We included the following sentences in the discussion: 

“Hcy levels of participants might be influenced by the supplementary intake of vitamins. We were not able to account for vitamin status, intake of supplementary vitamins, and a history of hyperhomocysteinemia treatment in our analysis, because the data has not been collected.”

I do not think figure S1 is necessary to show.

I would prefer to keep this supplementary figure to illustrate and justify our selection of confounders we included in our regression. In addition, one of the previous reviewers requested to be more transparent regarding the confounder selection and the justification of confounder selection.

Discussion: It would be of interest to discuss a the potential molecular basis of Hcy and myopia (e.g. LINE-1 hypermethylation), next to the limitations of the study. Additionally, it would be of interest the discuss the present data in the context of literature. Why do the authors think, that there is no association of Hcy and myopia – as there is evidence in literature for this association? 

To our knowledge, we are the first to hypothesize and explore a potential association of moderate elevated Hcy and refraction. This hypothesis is based on the established strong association between severe hyperhomocysteinemia and myopia, as we stated in the introduction (line 86). We added the following sentences to the discussion: 

“To our knowledge, this is the first study that has evaluated a potential relationship of Hcy and myopia. We assumed that moderately increased Hcy would be present over the entire life span and could thus be detected in our adult study sample. Given the proposed pathophysiological mechanism underlying the association of severe hyperhomocysteinemia and progressive myopia, namely that the extremely high Hcy concentration disrupts the zonular fibers, long-term exposure to moderately increased Hcy might also result in ocular effects like a more myopic refraction compared to the general population, and/or a continuous change towards more myopic refraction. This effect might either be observable in a cross-sectionally or in change over time – but we did not observe any effect in our study sample.”

The GHS is a great study with an even large cohort of participants, thus the data are of great interest – yet, some minor points should be considered and the discussion could be improved.

Thank you!

---

## [Editor Report · Decision Letter 1]

16 Mar 2020

No evidence for an association of plasma homocysteine levels and refractive error – Results from the population-based Gutenberg Health Study (GHS)

PONE-D-19-25431R1

Dear Dr. Nickels,

We are pleased to inform you that your manuscript has been judged scientifically suitable for publication and will be formally accepted for publication once it complies with all outstanding technical requirements.

With kind regards,

Sabine Rohrmann

Academic Editor

PLOS ONE
---

## [Editor Report · Acceptance letter]

27 Mar 2020

PONE-D-19-25431R1 

No evidence for an association of plasma homocysteine levels and refractive error – Results from the population-based Gutenberg Health Study (GHS) 

Dear Dr. Nickels:

I am pleased to inform you that your manuscript has been deemed suitable for publication in PLOS ONE. Congratulations! Your manuscript is now with our production department. 

With kind regards,

on behalf of

Dr. Sabine Rohrmann 

Academic Editor

PLOS ONE